# DiffMaSIF: Score-Based Diffusion Models for Protein Surfaces

## Abstract

Predicting protein-protein complexes is a central challenge of computational structural biology. However, existing state-of-the-art methods rely on co-evolution learned on large amino acid sequence datasets and thus often fall short on both transient and engineered interfaces (which are of particular interest in therapeutic applications) where co-evolutionary signals are absent or minimal. To address this, we introduce DiffMaSIF, a novel, score-based diffusion model for rigid protein-protein docking. Instead of sequence-based features, DiffMaSIF uses a protein molecular surface-based encoder-decoder architecture trained via a novel combination of geometric pre-training tasks to effectively learn physical complementarity. The encoder uses learned geometric features extracted from protein surface point clouds as well as geometrically pre-trained residue embeddings pooled to the surface. It directly learns binding site complementary through prediction of contact sites as both pretraining and auxiliary loss, and also allows for specification of known binding sites during inference. It is followed by a decoder predicting rotation and translation via $SO(3)$ diffusion. We show that DiffMaSIF achieves SOTA among Deep Learning methods for rigid body docking, in particular on structurally novel interfaces and low sequence conservation. This provides a significant advance towards accurate modelling of protein interactions with low co-evolution and their many practical applications.

## 1 Introduction

Proteins orchestrate many cellular functions, many of which are derived from the way in which they mutually interact. Protein 3D structure defines its function and interactions with other molecules. Recent groundbreaking work (Jumper et al. (2021)) showed that deep learning methods could be used to predict a significant fraction of protein structures to near-experimental accuracy using the protein sequence and information about its evolutionary history. The accurate prediction of protein-protein interactions, however, still remains an open challenge (Ozden et al. (2023)).

Traditionally, protein interaction is structurally determined through *docking*, where one attempts to predict the conformations of proteins in the complex from the individual structures of the interacting proteins. Protein-protein docking methods typically involve constructing a pseudo-energy function derived from physical principles fitted on known protein-protein complexes, potentially combined with known templates and heuristics (Vajda & Kozakov (2009)). Black-box stochastic optimization techniques are then used to search for minima within the energy functions. However, the search space of all possible conformations including backbone and side chain torsions is infeasible to explore exhaustively (Harmalkar & Gray (2021)), therefore sampling techniques such as Monte Carlo simulations are applied (Marze et al. (2018)). As an initial approximation, *rigid-body docking* (where the relative pose of one protein with respect to the other is determined) is often performed, sometimes followed by an iterative refinement allowing backbones and side chains to relax in presence of its interacting partner (Desta et al. (2020)).

Current deep learning methods for protein-protein docking typically build on the same principles as structure prediction, leveraging sequence representations trained via masked-language modelling on large evolutionary sequence databases such as UniRef (Ketata et al. (2023); Jumper et al. (2021)). While these tend to perform well in case of co-evolved stable interfaces, they fail to capture the many structurally diverse, transient and flexible interactions many proteins participate in. In addition,

**Surface Encoding**   **Auxiliary Task**   **Diffusion**

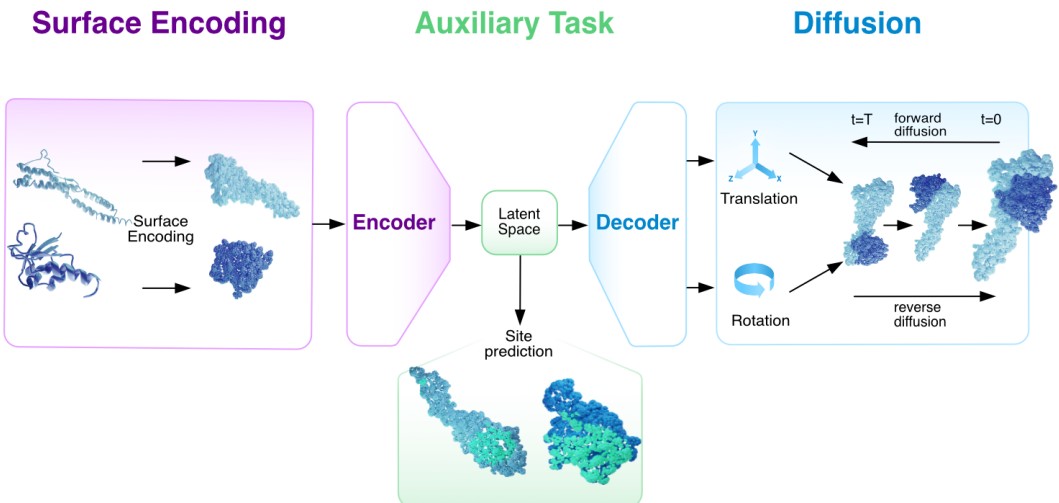

Figure 1: Overview of the DIFFMASIF method. Protein surface point clouds are generated and fed into a encoder-decoder network. The model both learns binding site prediction via encoder and denoising a reverse diffusion process over rotations and translations via decoder

de-novo designed interfaces as well as heavily recombined sequences such as antibody hypervariable regions, which are very commonly used for therapeutic applications, lack co-evolution data. This leads to subpar performance of existing deep learning approaches (Ozden et al. (2023)).

It is known, however, that all protein interactions are mediated and understandable through steric and electrostatic complementarity of the interface (Lawrence & Colman (1993); Jones & Thornton (1996)). Early rigid-body docking approaches (Katchalski-Katzir et al. (1992)) in fact relied on implicit representations of protein surfaces and by using fast Fourier transform of a correlation function to assess the degree of shape complementarity. Later, in a deep learning context, learned protein surface representations (*molecular surface interaction fingerprinting*, or *MaSIF*), which can capture this steric and electrostatic complementarity have proven to be powerful in predicting protein interactions Gainza et al. (2020); Sverrisson et al. (2021); Gainza et al. (2023).

In this paper, as a way to address the limitations of co-evolution based approaches, we propose DIFFMASIF, the first score-based diffusion model for rigid-body docking using a versatile surface representation of proteins.

**Main contributions.** DIFFMASIF is the first protein surface-based diffusion model, addressing the limitations of current co-evolution reliant models. Second, we propose a novel pre-training scheme comprising two geometric pre-training steps to replace and surpass evolutionary pre-trained sequence embeddings. Third, we show that DIFFMASIF has the ability to include one- or two-sided binding information (if available) or directly predict it at inference time. Fourth, we use a novel encoder-decoder architecture that combines a surface-pooled vector-neuron (DGCNN) encoder with E(3)-equivariant graph convolution decoder, trained to learn binding site structural complementarity and rigid body docking via SO(3) diffusion, respectively. Finally, we show state-of-the-art rigid body docking results, surpassing current machine learning methods on structurally novel interfaces.

## 2 BACKGROUND

### 2.1 DEEP-LEARNING BASED METHODS FOR PROTEIN DOCKING

EquiDock (Ganea et al. (2021)) is a deep learning method for rigid-body protein-protein docking, forming a graph representation of the residues in the interacting proteins, and predicting (through SE(3)-equivariant operations) keypoints for their alignment using the Kabsch algorithm. EquiBind (Stärk et al. (2022)) was later developed as an extension of EquiDock for the docking of small molecules to proteins. AlphaFold-Multimer (Evans et al. (2021)) is an extension of AlphaFold2 (Jumper et al. (2021)) for protein-protein complexes.

Both AlphaFold-Multimer and EquiDock are trained discriminatively. Generative models provide an advantage here as they can the ensemble of bound protein confirmations. One recent example is DiffDock (Corso et al. (2022)), a diffusion generative model for docking small molecules to proteins. A derivative, DiffDock-PP, (Ketata et al. (2023)) was developed for predicting rigid-body protein-protein docking via SO(3) diffusion on rotations and translations.

The DIFFMASIF method proposed in this paper also learns a denoising process on the space of translational and rotational degrees of freedom, inspired by DiffDock-PP. However, we make significant contributions to address limitations of DiffDock-PP and other co-evolution reliant models by exploiting surface-based representation (Gainza et al. (2020)). To our knowledge, DIFFMASIF is the first surface-based generative diffusion model for protein docking. We also introduce a novel architecture and dedicated pre-training-fine-tuning setup to learn structural complementarity rather than sequence homology or co-evolution. Furthermore, we incorporate knowledge-based priors that are often available at inference time to augment the featurization.

## 2.2 SCORE-BASED DIFFUSION MODELS

Score-based diffusion models integrate techniques from both score-based generative models and diffusion models into a unified framework (Song & Ermon). In score-based generative modeling, the score function $\mathbf{s}_\theta(\mathbf{x}) \approx \nabla_\mathbf{x} \log p(\mathbf{x})$ represents gradients of the data log-density $p(\mathbf{x})$. The score can be estimated via denoising score matching on noise-corrupted samples, without needing to compute intractable normalizing constants. Langevin dynamics can then sample from the estimated score model. Diffusion models perturb data $\mathbf{x}_0$ through Markov chains of added Gaussian noise to obtain $\mathbf{x}_t$ at noise level $t$. The forward diffusion process can be represented as a stochastic differential equation (SDE):

$$d\mathbf{x} = f(t)dt + g(t)d\mathbf{w} \tag{1}$$

where $f(t)$ and $g(t)$ represent drift and diffusion coefficients respectively, and $d\mathbf{w}$ is Gaussian noise. The reverse process is modeled by learning an approximate conditional distribution $p_\theta(\mathbf{x}_{t-1}|\mathbf{x}_t)$. Score-based diffusion models leverage score functions to parameterize the generative diffusion process. The forward SDE incrementally adds noise to the data distribution $p_0(\mathbf{x})$. Critically, the reverse-time SDE is:

$$d\mathbf{x} = [f(t) - g(t)^2 \nabla_\mathbf{x} \log p_t(\mathbf{x})]dt + g(t)d\mathbf{w} \tag{2}$$

The score functions $\nabla_\mathbf{x} \log p_t(\mathbf{x})$ can be estimated by a time-dependent score-based model $\mathbf{s}_\theta(\mathbf{x}, t)$ trained via score matching. This results in an estimated reverse SDE that can be numerically solved to sample from $p_0(\mathbf{x})$. Alternatively, the estimated reverse SDE can be converted to a probability flow ODE, enabling exact likelihood computation (Song et al.)

## 2.3 DEEP LEARNING ON PROTEIN SURFACES

MaSIF (Gainza et al. (2020)) is a geometric Riemannian convolutional architecture (Monti et al. (2017)) used to learn protein surface descriptors for predicting interaction properties. MaSIF operates using local geodesic patch operators (a discrete version of an exponential map on a Riemannian manifold) on solvent-accessible protein molecular surfaces represented as meshes. In (Gainza et al. (2023)), it was shown that MaSIF could be used to design *de novo* protein interactions.

In its faster and end-to-end differentiable successor dMaSIF (Sverrisson et al. (2021)), points are generated in an iterative procedure, where for a sampled set of points $x_i \in X$ their distance to a level set of a signed distance function ($SDF$) is minimized, leading to surface points roughly equidistant from each surface atom. For each point, a local coordinate system $\hat{n}_i, \hat{u}_i, \hat{v}_i \in \mathbb{R}^3$ is constructed using the gradient of the distance function and subsequent calculation of tangent vectors. This local coordinate system is then used to define quasi-geodesic convolutions along the surface, which are

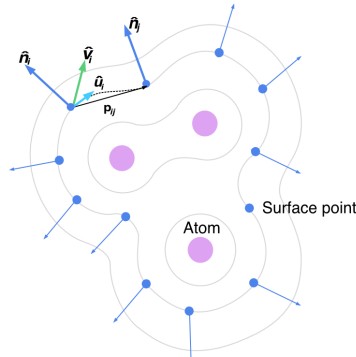

Figure 2: dMaSIF surface point cloud featurization. Level sets (gray) around atoms (pink) are used to find surface points (blue). Each point is equipped with a coordinate system comprised of normals $\hat{n}_i$ (blue) and orthogonal tangent vectors $\hat{v}_i, \hat{u}_i$.

used to propagate a 16-dimensional feature vector for each point. The 16 input features of each point are comprised of 10 geometric descriptors (mean- and Gaussian curvatures at 5 different scales) and a 6-dimensional vector stemming from its 16 nearest atoms. Each atom is one-hot encoded by their respective atom types $[C, H, O, N, S, Se]$ and concatenated to the atom-point distance, creating an embedding in $\mathbb{R}^7$ which is pooled to the closest point via an MLP. With DIFFMASIF we further build on these representations.

## 3 DATA

The typical benchmark set in protein docking is derived from the Dataset of Interacting ProteinS (DIPS) (Townshend et al. (2019)), with a test split comprised of Docking Benchark 5 (DB5)(Vreven et al. (2015)) after removal of all proteins with more then 30% sequence homology. However, DB5 is a small test set compared to DIPS training set size. Equally, sequence similarity is often not a good predictor of structure similarity, with some proteins having high global sequence similarity but structural differences at the interface.

To address these issues we introduce new splits intended for rigid-body docking by using a structural interface clustering approach. This is consistent with our desire to reflect the performance of current methods by their ability to learn structural complementarity and generalize towards novel (potentially weakly co-evolved) interfaces. The splits were prepared as follows:

We retrieved all protein complexes from the PDB (cutoff date: March 2023) without resolution cutoff, and performed all-vs-all structural alignments of all available chains using Foldseek (van Kempen et al. (2023)). While storing local alignment positions, Foldseek normalizes the alignment scores as TM-score, which is used to filter out alignments with lower structural similarity (TM-score $< 0.6$). We classified residues as binding site residues based on an 6Å heavy chain atom distance threshold between chains, and a pair of chains was defined as interacting if there were at least 6 binding site residues, and at least 50% of these were covered by the Foldseek alignment. We created a graph of interacting chain pairs with the interface alignment TM-scores as edge weights and performed community clustering to define interface clusters. We then sampled 10% of the clusters as test set, which contained at least one high quality representative PPI. We filtered by dimeric state, experimental method (X-ray), resolution (4.5A), more than 5 atom types, no missing residues at interface and ensuring chain lengths between 25 and 550 residues.

We retain the highest resolution among each clusters as test set. We further inspected the original data and separated the representative structures with a few gaps at non-interfacial locations to form a validation set of 190 members, with ungapped high quality representatives used as the test set with 505 diverse dimers in total. The remainder is filtered by structures with more than 5 atom types, arriving at 119,562 train complexes.

To estimate the level of co-evolutionary signals present in our test-set we ran the multiple sequence alignment (MSA) creation step of ColabFold (Mirdita et al.) which generates an MSA of the MMSeqs2 hits from UniRef100, PDB70 and an environmental sequence set. These MSAs were used to calculate the average number of effective sequences ($N_{\text{eff}}$) at each position, a measure of availability of homologous sequences that can be utilized by evolution-based approaches.

In addition, to compare DIFFMASIF results with those of AlphaFold-Multimer, we curated a set of 200 dimers released after 01.01.2023 which do not have any Foldseek hits to any of our interface clusters. This is referred to as the AF-Multimer-set.

## 4 METHODS

### 4.1 DIFFUSION PROCESS

Our study adopts a diffusion process approach, akin to the methodologies presented in Corso et al. (2022); Ketata et al. (2023). We focus on a combined space termed the product manifold, denoted as $\mathbb{P}$. This manifold is a combination of:

1. **3D Translation Group** $\mathbb{T}(3)$: Essentially, this is the space of all possible 3D translations, equivalently represented as $\mathbb{R}^3$.

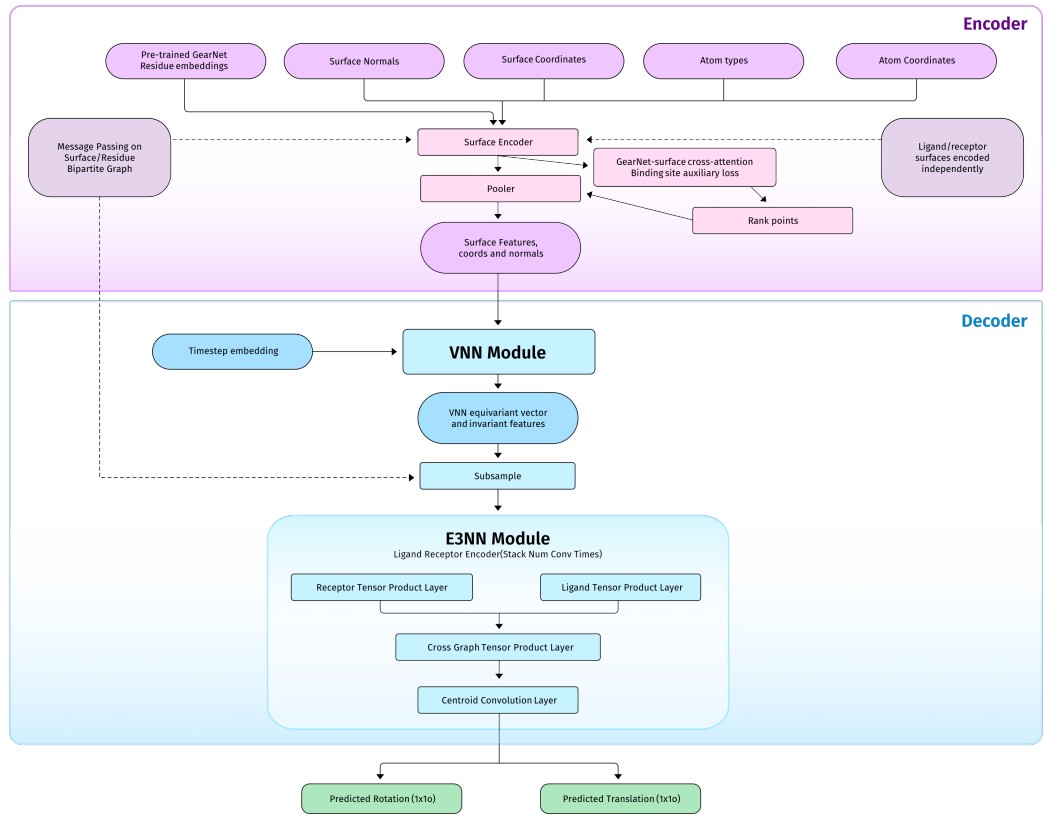

Figure 3: The DiffMaSIF architecture is constructed as encoder-decoder to learn different aspects of the problem, respectively. The encoder utilizes surface-level representations extracted from surface point clouds along with their coordinates and normals. It additionally leverages geometrically pretrained residue embeddings pooled to the surface. The decoder use DGCNN (vector neuron) layers and an E(3)-equivariant graph convolution layer to predict translation and rotation scores.

2. **3D Rotation Group** $SO(3)$: This encapsulates all conceivable 3D rotations.

**Translations**: For any translational movement in the 3D space, we employ the equation:

$$d\mathbf{x} = \sqrt{d\sigma_{tr}^2(t)/dt}\, d\mathbf{w}$$

Where: - $d\mathbf{x}$ signifies the change in position. - $\sigma_{tr}^2(t)$ is the diffusing variance at a specific time $t$. - $d\mathbf{w}$ represents the 3D Brownian motion, a type of random motion in three dimensions.

**Rotations**: For the rotational aspect, the process is twofold:

1. We initially select a random axis, represented as $\hat{\omega}$, and a random angle $\omega$ constrained between 0 and $\pi$. 2. The likelihood of opting for a particular angle $\omega$ is expressed by:

$$p(\omega) = \frac{1 - \cos\omega}{\pi} f(\omega)$$

Where $f(\omega)$ is a truncated series expression, as detailed in Leach et al. (2022).

By distinctly defining the diffusion for $\mathbb{T}(3)$ and $SO(3)$, we can train a model to match the scores for each kind of movement. During the sampling phase, we amalgamate samples from both the translational and rotational processes. This involves a random rotation of the ligand around its centroid and a random translation. This integrated methodology facilitates the creation of a generative model for docking within the product manifold $\mathbb{P}$.

## 4.2 MODEL ARCHITECTURE

The architecure of DIFFMASIF has two main components: an encoder and a decoder. The encoder utilizes surface-level representations extracted from surface point clouds along with their coordinates and normals. It additionally leverages geometrically pretrained residue embeddings pooled to the surface. The decoder use DGCNN (vector neuron) layers and an E(3)-equivariant graph convolution layer to predict translation and rotation scores.

**Input features.** DIFFMASIF takes residue and atom level features, later pooled to the surface, as input for both proteins. We refer to the chain whose coordinates are updated as the ligand and the stationary chain as the receptor. To focus our model on structure-derived signals, we made use of pretrained GearNet embeddings (Zhang et al. (2022)) as scalar input features for residues, along with their coordinates. The atom level features, which consist only of heavy atoms, contain one hot encoding of atom types and coordinates, which are passed to a dMaSIF layer to generate surface normals and positions along with scalar embeddings from dMaSIF's geodesic convolution layer. Then we scale both GearNet features and dMaSIF scalar embeddings to the same dimensions using MLP layers.

**Encoder.** DIFFMASIF first constructs a heterograph $G = (R, S, \mathbf{E}_{SS}, \mathbf{E}_{RR}, \mathbf{E}_{SR})$ composed of residue nodes $R$, surface nodes $S$, intra-representation edges $\mathbf{E}_{RR}$ and $\mathbf{E}_{SS}$ and inter-representation edges $\mathbf{E}_{SR}$. These edges are created using a radius graph between each node type. The heterographs are constructed for the protein and ligand separately and therefore no cross information between ligand and receptor is communicated at this stage. We stack several layers of EGNN (Satorras et al.) computed over intra-representation graphs $(S, \mathbf{E}_{SS}), (R, \mathbf{E}_{RR})$ and inter-representation graphs $(S, R, \mathbf{E}_{SR})$ to obtain our equivariant surface scalar features.

**Binding-Site Auxiliary Task.** In DockGPT (McPartlon & Xu), it was shown that including only one contact point during generation greatly improves complex prediction. This led us to hypothesize that, while related, binding-site prediction and pose generation are fundamentally distinct tasks. Therefore we sought to construct a loss that differentiates interaction site prediction from pose prediction. This binding site auxiliary loss passes the result of the residue-to-surface message passing network from both the ligand and the receptor through a cross-attention mechanism to predict whether a surface node is part of the binding site or not. True binding site nodes are defined as those $< 3$Å from the other surface. Only the top 512 predicted binding site nodes are used from the ligand and receptor for the decoder, reducing the high compute and memory required by the decoders tensor-product convolution layers. The combined denoising score loss and auxiliary binding loss is:

$$\mathscr{L} = \lambda \text{BCELoss}(\hat{c}, c) + S(s_\theta(x(t), \text{argmax}_{512}(\hat{c})), \phi, \psi),$$

where $\hat{c}$ and $c$ are the predicted contact probabilities and ground truth contacts respectively, $S$ is the denoising score loss used in DiffDock-PP, $s_\theta$ is the decoder model and $\phi$ and $\psi$ are the true rotation and translation scores sampled at time step $t$ from $p(x_\phi(t)|x_\phi(0))$ and $p(x_\psi(t)|x_\psi(0))$. Instead of the full noise perturbed coordinates, the decoder model receives only perturbed coordinates at time step $x(t)$ masked to only include the 512 most likely contacts. We balance these two losses with $\lambda$, a weighting term.

**Decoder.** The decoder works on the joint PPI graph consisting of the top 512 predicted binding site nodes of both the ligand and the receptor. The first component of the decoder is a DCGNN (vector neuron layers) (Wang et al. (2019); Deng et al. (2021)) which takes as input coordinates and normal vectors and outputs higher-dimensional vector embeddings. These vector features are then provided, along with surface coordinates, and surface scalar features to an E(3)-equivariant graph convolution layer constructed using the E3nn library (Geiger & Smidt (2022)). The final output of the decoder is the prediction of the translation and rotation required for the ligand coordinates. The combination of DGCNN and E(3)-equivariant graph convolutions ensure the process respects the geometric constraints of the protein surface structure and enhances the model's ability to predict complex protein interactions.

## 5 RESULTS

### 5.1 DIFFMASIF LEARNS RELEVANT PROTEIN-PROTEIN COMPLEX CHARACTERISTICS

**Binding site prediction.**   Using the dMaSIF contact probabilities, the DIFFMASIF encoder selects the 512 most probable contacts for both the receptor and the ligand to use in further steps for pose prediction. This ranking is based on the binding site auxiliary loss and thus is trained to cover the interfaces of both proteins. To evaluate the accuracy of binding site prediction, we evaluated the trained model on 250 randomly selected complexes from the test set. We show that from the 512 top ranked DIFFMASIF surface points, 50% lie directly at the correct interface, a significant improvement over randomly sampling 512 surface points via random pooling, which would result in merely 12% of interface sites. Note that sampling 100% of the 512 points from the binding site is unlikely to be desirable as the model might want to leverage reference points not directly at the binding site to avoid clashing of two proteins at peripheral sites. Figure 4A shows the distribution of precision and recall binding site sampling of DIFFMASIF during inference compared to random sampling of the equivalent 512 surface points. Figure 4B shows how interface RMSD improves with increasing binding site coverage. Thus, DIFFMASIF learns to recognize binding surfaces thanks to the cross-attention between learned surface embeddings. This two-step scheme also has the added benefit that known binding site information from either or both proteins can be utilized at inference time to further improve interface specificity.

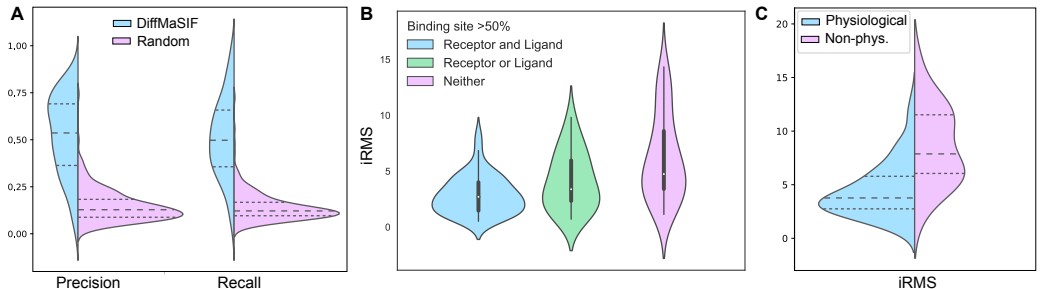

Figure 4: **A**) Precision and recall of DIFFMASIF binding site prediction compared to random sampling **B**) Interface RMSD distributions of complexes with accurate binding site prediction, i.e >50% of the predicted surface points lie in the interface, for both ligand and receptor, either one of ligand or receptor, and neither. **C**) Interface RMSD distributions of physiological and non-physiological (crystal artifact) complexes.

**Physiological interface prediction.**   As deep learning methods are often seen to be biased due to the inherent biases in their training data , we wanted a complementary approach to verify that DIFFMASIF learns biologically relevant surface and structural complementarity signals. We demonstrate this is Figure 4B by comparing the iRMSD distributions of 84 physiological interfaces vs. 20 interfaces resulting from crystal artefacts (non-physiological), as determined by PRODIGY (Vangone & Bonvin). The differing distributions clearly favour physiological interfaces and confirms that DIFFMASIF learns structural complementarity without overfitting, despite the unavoidable levels of noise present in protein complex experimental structure data.

### 5.2 DIFFMASIF OUTPERFORMS CO-EVOLUTION BASED DOCKING METHODS

We benchmarked DIFFMASIF against popular deep learning-based methods using the test subset of our dataset, as described in 3. As it is difficult to retrain the AF2MM co-folding method with our data splits, we compare to this method using a different hold-out set (AF2MM-set) described in Section 3. Each pose generated by a particular docking method was superposed to the reference pose and evaluated based on three metrics: iRMSD (interface RMSD), LRMSD (ligand RMSD), and DockQ (a composite score that also encompasses Fnat – the fraction of native contacts). For DIFFMASIF, we generated 40 poses per complex with a reverse ODE using 40 steps. For all complexes, we take the pose with the best valued metric when computing the statistics below (oracle metrics).

| Method | Interface RMSD | Ligand RMSD | Fnat | DockQ | CAPRI |
|---|---|---|---|---|---|
| | Median | Median | Median | Median | Hit Rate (%) |
| EquiDock | 17.64 | 34.139 | 0.0 | 0.02 | 0 |
| DiffDock-PP 1 | 15.29 | 34.13 | 0.0 | 0.029 | 5.54 |
| DiffDock-PP 10 | 6.88 | 16.13 | 0.13 | 0.12 | 33.8 |
| DiffDock-PP 40 | 4.34 | 10.53 | 0.29 | 0.26 | 54.9 |
| AF2MM* | 8.285 | 14.449 | 0.055 | 0.138 | 49.2 |
| DIFFMASIF 1 | 13.74 | 30.24 | 0.01 | 0.04 | 5.0 |
| DIFFMASIF 10 | 5.86 | 14.57 | 0.19 | 0.16 | 37.4 |
| DIFFMASIF 40 | **3.35** | **8.63** | **0.37** | **0.34** | **64.3** |

Table 1: Benchmark results comparing complex prediction by EquiDock, DiffDock-PP and DIFF-MASIF on the test set and AF2MM on the AF2MM-set. Number next to method give number of poses oracle metrics are generated from for methods with variable number of poses. CAPRI Hit Rate is defined as percentage of systems with acceptable or better solutions

**DiffDock-PP,** We retrained DiffDock-PP on our novel data splits, after having ensured reproducibility of results previously reported using the original splits, and generated 40 poses per test complex with a reverse SDE using 40 steps. As seen in Table 1, DIFFMASIF shows consistently better scores than DiffDock-PP and returns more acceptable complexes.

In addition, Figure 5A demonstrates the percentage of acceptable poses returned by DiffDock-PP and DIFFMASIF as a function of the average number of effective sequences ($N_{\text{eff}}$) available per complex. This can be seen as a measure of co-evolution, as complexes with a high detectable co-evolutionary signal would be expected to have deep MSAs. Both DiffDock-PP as well as co-folding methods such as AlphaFold-Multimer include representations learned via masked language modelling objective on this information. From Figure 5A, it is clear that DiffDock-PP performance drops when the average $N_{\text{eff}}$ goes below 100, while DIFFMASIF is not effected. Thus, DIFFMASIF is a highly complementary approach to rigid protein docking without reliance on co-evolutionary signals. Figure 6 shows some picked examples showcasing the DIFFMASIF successes in comparison to DiffDock-PP.

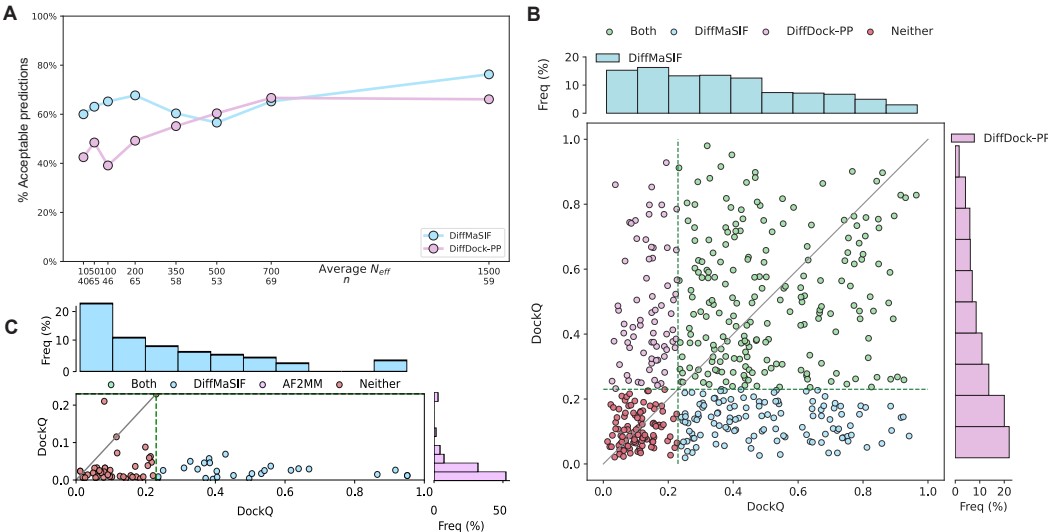

Figure 5: **A**) Percentage of acceptable DiffDock-PP and DIFFMASIF complexes across different $N_{\text{eff}}$. **B**) DiffMaSIF and DiffDock-PP DockQ distributions. The size of each data point corresponds to the number of intermolecular contacts in the complex. **C**) DockQ scores of complexes returned by AF-Multimer and DIFFMASIF

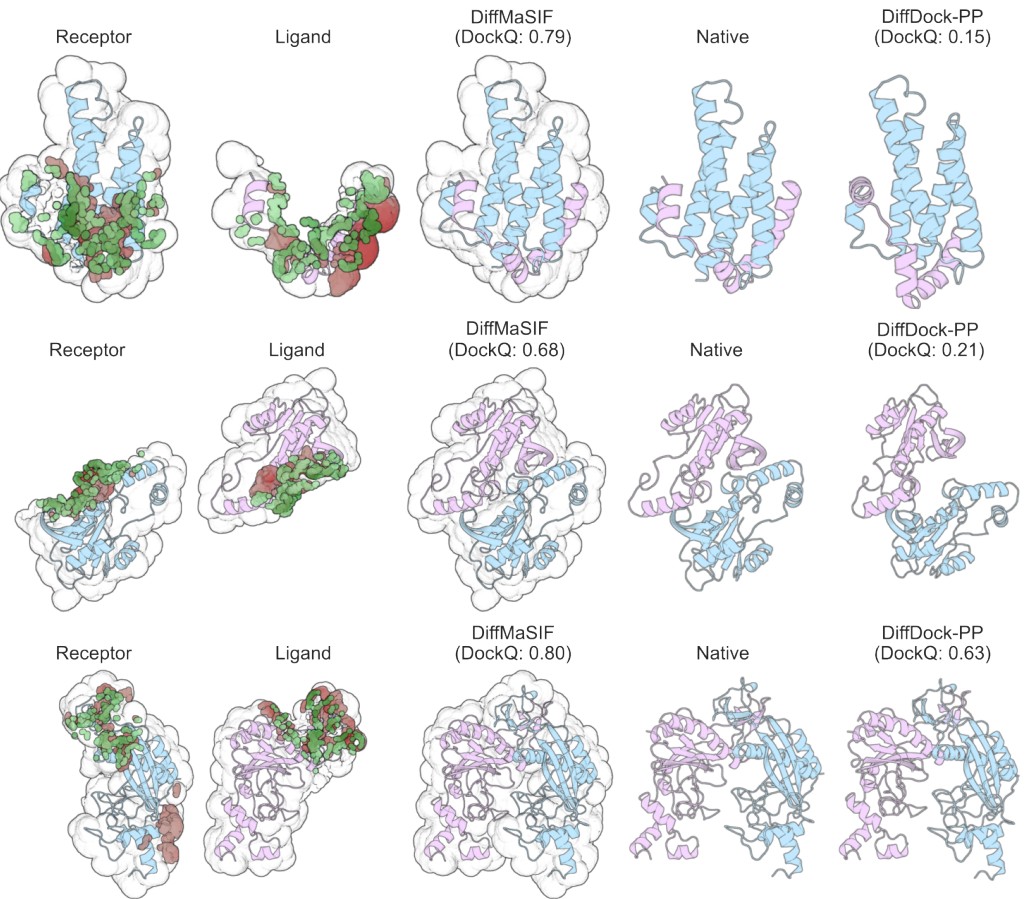

Figure 6: **A)** Docking of PDB ID: 1ORY, 2BZ0 and 1RVE. The two leftmost columns show DIFF-MASIF abilities to correctly identify binding sites, with correctly identified interface points in green, and incorrectly shown in red, indicating high concordance. The last three rows show docking performance with ground truth in the fourth column. As can be seen, DIFFMASIF is able to rescue instances where DiffDock-PP fails to produce acceptable poses

**Alphafold2-Multimer.** Unlike the rigid body docking methods described here, AlphaFold2-Multimer (AF2MM) (Evans et al. (2021)) takes the sequences of both proteins as input and co-folds them into a complex structure, with a heavy reliance on co-evolutionary signals between the interfaces involved. We ran the default AF2MM pipeline for the AF2MM-set of 200 complexes (3), with the exception of removing template structures dating September 2022 and onward. This resulted in 5 models per complex from which we took the model with the highest DockQ score, i.e an oracle approach. AF2MM returned unacceptable (DockQ $< 0.23$) models for 103 of these complexes. DIFFMASIF predictions were acceptable or better for 38% of these (Figure 5C), again demonstrating the complementarity of our approach for difficult interfaces.

# 6 CONCLUSION

We present the first purely structure- and surface-based deep learning model for protein-protein docking. Our results demonstrate that ML models can achieve comparable results without the use of co-evolutionary information, and out-perform in situations where such information is scarce or not expected. This effort expands our toolbox for leveraging physico-electrochemical surface characteristics of proteins and lends well to future efforts where the right combination of co-evolution and structural complementarity can be learned across protein-protein space. In addition, we demonstrate the power of learning joint interface prediction and pose generation, also enabling the use of knowledge-based priors to improve prediction specificity. Overall, our surface point graph and atom-to-surface pooling approach represent a step forward for protein representation learning especially in the context of generative modeling.

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
