# OpenReview forum: "DiffMaSIF: Score-Based Diffusion Models for Protein Surfaces"
_ICLR.cc/2024/Conference — Submitted to ICLR 2024_

### Official Review · Reviewer_z8Wu · 2023-10-19

**Soundness:** 2 fair
**Presentation:** 1 poor
**Contribution:** 2 fair
**Rating:** 3
**Confidence:** 4

**Summary:**

The paper studies the problem of rigid-body protein-protein docking. In this context, given two protein chains, the goal is to predict the bound complex structure without accounting for the transformation of the protein structures. The introduced method, DiffMaSIF, employs a surface-based encoder to predict the binding site. Subsequently, these learned representations are utilized to determine the score function within SO(3) diffusion. The authors demonstrate DiffMaSIF's effectiveness by contrasting it with other deep learning approaches on a newly established benchmark. Additionally, they highlight the method's efficacy on proteins with limited MSA depths.

**Strengths:**

1. Protein docking is an important problem in protein science. Utilizing surface-based techniques appears suitable for pinpointing the physical complementarity between ligands and binders. Such an approach might circumvent the generalization issues frequently encountered in many deep learning methods due to overfitting.
2. Experimental results for proteins with low-depth MSAs are encouraging. These findings suggest that surface-based methods are less dependent on co-evolutionary information compared to other techniques.

**Weaknesses:**

1. The paper omits many important details regarding the proposed method and experiments, which hampers comprehension and replication of the method.
2. The techniques employed in the paper are primarily borrowed from existing literature, making it lack of novelty.
3. The authors introduce a new dataset for evaluating the methods. However, it is unclear why this new dataset based on structural similarity is more appropriate. According to the statement “Equally, sequence similarity is often not a good predictor of structure similarity, with some proteins having high global sequence similarity but structural differences at the interface”, the datasets like DIPS and DB5, split by sequence similarity, present greater generalization challenges.
4. Many important baselines are missing in the new benchmark. The comparison made in the experiments is not fair.

For details, please refer to the Question section.

**Questions:**

Major points:
1. The title seems inaccurate. While the paper introduces a diffusion model in SO(3) space, it does not directly work on protein surfaces. The term "protein docking" should be incorporated to avoid misleading readers into interpreting it as a generative model for designing protein surfaces.
2. The paper is missing a 'Related Work' section that would establish context with existing studies. There is a wealth of related research on traditional protein docking, protein representation learning, and diffusion models for proteins. Furthermore, the discussion in Sec. 2 is brief. It would benefit from a more in-depth comparison of DiffMaSIF's advantages over existing methods like DiffDock-PP, AlphaFold-Multimer, and MaSIF.
3. The paper omits vital details about constructing the heterograph in the encoder and the joint PPI graph in the decoder — both of which are central to the method.
4. The design of the cross-attention between ligands and receptors for binding site prediction remains unclear.
5. In Figure 4A, what criteria are used to determine if two surface nodes bind? Conventionally, a precision-recall curve is plotted instead of presenting precision and recall values across all complexes.
6. The paper should include comparisons with deep learning benchmarks such as MaSIF and DiffDock-PP in Figure 4, especially since the authors argue that "deep learning methods often exhibit biases stemming from their training data."
7. Table 1 seems to miss several baselines for rigid-body docking, such as HDOCK[1], ClusPro[2], and PatchDock[3]. To highlight the effectiveness of diffusion models, results from MaSIF should also be included.
8. Leveraging an oracle to select optimal results evidently overestimates DiffMaSIF's performance. A more appropriate approach would involve training a confidence model to choose the best prediction, similar to DiffDock's strategy.
9. The comparison with AF2MM in Table 1 seems unfair. How can the results be comparable when the test set for AF2MM differs from the other methods?

Minor points:
1. Figure 1: The design of the figure needs improvement. The three subtitles appear disproportionately large, while the other text elements are notably small. The term "auxiliary task" is positioned at the top, yet its corresponding green box is at the bottom. Additionally, the protein structure within the blue box overextends beyond its boundary. Similar issues for Figure 2.
2. Sec. 3 and 4 would benefit from a switch in order. It's more logical to discuss the method after laying out the basic knowledge and to relocate the dataset discussion to the experiment section.
3. In the fourth paragraph of Sec. 3, the phrase "separated the representative structures with a few gaps at non-interfacial locations" is ambiguous. Where do these "gaps at non-interfacial locations" come from?
4. Sec. 4.1: Although it is not difficult to infer that the translation and rotation group defines the transformation on ligands with domain knowledge, there's a need to formally define the docking problem and diffusion process.
5. Sec. 4.1: The definition of the diffusion process on the rotation group remains unclear. The equation provided describes a distribution within the $IG_{SO(3)}$ space but lacks clarity on how noise increases as time $t$ progresses. A more thorough explanation is needed here, rather than merely replicating an equation from prior literature.
6. Sec. 4.2: Expand on the denoising score loss, rather than merely referencing its definition in DiffDock-PP.
7. Figure 4: In A and C, labels are displayed on the x-axis. However, in section B, the label is oddly placed on the y-axis.
8. Table 1: The significant figures for different methods should be consistent.
9. Figure 5: (1) The caption indicates, "The size of each data point corresponds to the number of intermolecular contacts in the complex", yet all points appear uniform in size. (2) The criteria used to plot the green dashed lines needs clarification.
10. The citations within the paper exhibit formatting inconsistencies. Most references lack the publication year, and many solely cite the arxiv version.

Typos:
1. Page 3, Sec. 2.1: “as they can the ensemble of bound protein confirmations” -> remove “the” and “of”.
2. Page 3, Sec. 2.3: “In (Gainza et al. (2023))” -> use \citet instead of \citep for references appeared in sentences.
3. Page 4, Sec. 3: “We retain the highest resolution among each clusters as test set”, clusters -> cluster.
4. Page 6, Sec. 4.2, Binding-Site Auxiliary Task: $p(x_{\psi}(t)|x_{\psi}(0)$, lacks a right parenthesis.
5. Page 7, Sec. 5.1, Physiological interface prediction: Figure 4B -> Figure4C

Overall, I believe there's significant potential in exploring more applications of surface-based methods within the realm of protein docking tasks. That being said, the current version of the paper falls short in providing comprehensive details and experimental data, making it challenging to deem it ready for publication. I would strongly recommend the authors to reorganize and improve the paper to elevate its quality. This would not only solidify its scientific contributions but also make it a more compelling read for the community.

[1] Yan, Yumeng, et al. "The HDOCK server for integrated protein–protein docking." Nature protocols 15.5 (2020): 1829-1852.

[2] Kozakov, Dima, et al. "The ClusPro web server for protein–protein docking." Nature protocols 12.2 (2017): 255-278.

[3] Schneidman-Duhovny, Dina, et al. "PatchDock and SymmDock: servers for rigid and symmetric docking." Nucleic acids research 33.suppl_2 (2005): W363-W367

---

### Official Review · Reviewer_J4YD · 2023-10-30

**Soundness:** 3 good
**Presentation:** 3 good
**Contribution:** 3 good
**Rating:** 6
**Confidence:** 4

**Summary:**

The authors propose DiffMaSIF, the first surfaced-based diffusion model for rigid protein-protein docking. The encoder of DiffMaSIF takes surface- and residue-level inputs and extracts surface features, coordinates and normals. The decoder comprises a VNN module and an E3NN module to predict ligand roto-translation scores.
Experiments show that DiffMaSIF outperforms previous SOTA DL methods for rigid-body protein-protein docking, in particular on sequentially/structurally novel interfaces.

**Strengths:**

1. Novelty: This work is the first surface-based diffusion model for rigid protein-protein docking, providing a possible solution for interfaces with little co-evolution information available.
2. Significance: Significant performance improvement over the previous SOTA is claimed.
3. Good writing & figures: The methods section is well written (for audience familiar with the subject), and the figures are straight to the point and easy to understand (despite some confusion in the encoder part, Figure 3).

**Weaknesses:**

1. Insufficient experiments: To present DiffMaSIF in a more comprehensive way, I would argue that the runtime, # parameters and ablation study results shall be reported.

**Questions:**

1. Why are there no supplementary materials? Is it by mistake or is there no other information you would like to present?
2. Please explain the data splits more clearly. Currently there are three "as test set" phrases in the data section, which, at first glance, is really confusing. I suppose the first two are in fact "test set candidates" (to be filtered further) and the last one is the final test set. If possible, a supplementary figure explaining the data split pipeline would be very nice.
3. If AF-Multimer is evaluated on a separate test set, I would suggest adding a supplementary table showing the results of all models on the AF-Multimer-set, just for fair comparison.
4. Regarding comments in the weakness section, it would be best if you could provide (1) the runtime and # params of all models, (2) ablation study showing the importance of the residue-level features, (3) ablation study showing the rationality of the decoder design. Why is DCGNN necessary?
5. According to Figure 3, "subsample" happens after VNN. Is this step where the top 512 predicted binding site nodes are used for subsequent layers? If so, then it seems to contradict the description that "The decoder works on the joint PPI graph consisting of the top 512 predicted binding site nodes of both the ligand and the receptor." If not, then what does this step do?

---

### Official Review · Reviewer_mUha · 2023-10-31

**Soundness:** 3 good
**Presentation:** 2 fair
**Contribution:** 2 fair
**Rating:** 5
**Confidence:** 5

**Summary:**

The paper introduces DiffMaSIF, a unique score-based diffusion model tailored for rigid protein-protein docking, particularly in scenarios lacking co-evolutionary signals. Leveraging a protein molecular surface-based encoder-decoder architecture, DiffMaSIF excels in learning physical complementarity, showcasing superior performance compared to existing deep learning methodologies.

**Strengths:**

1. The paper presents a surface-based model rooted in the DiffDock-PP diffusion framework, specifically for protein-protein rigid docking. Experimental results affirm its superiority over DiffDock-PP, highlighting its effectiveness.
2. The proposed surface-based method can avoid the need for evolutionary pretrained sequence embedding.

**Weaknesses:**

1. The method is constrained to rigid docking scenarios, assuming the availability of holo-like structures. This limitation becomes apparent in real-world applications where only apo-structures are accessible, and the protein surface, particularly the binding site, may exhibit significant flexibility.
2. The paper lacks an ablation study, leaving questions about the impact of pretrained GearNet embedding and the role of different level representations unanswered.
3. The technical contribution is somewhat incremental. The work primarily builds upon the DiffDock-PP diffusion framework for docking pose prediction, with the introduction of a surface-based encoder, a concept previously explored in existing literature [1].
4. The method employs "oracle selection" to choose the best pose based on the highest valued metric, a practice that may not provide a fair comparison against end-to-end prediction models like AFM and EquiDock. While generating multiple poses is feasible, it typically necessitates an additional selection module, as seen in DiffDock, to enable the model to autonomously select the binding pose.
5. The paper does not provide results or discussion regarding the inference efficiency of the proposed method.
6. There is no provided code or anonymous link to facilitate the reproduction of the results, which could be a potential barrier for other researchers.

[1]. Sverrisson, Freyr, et al. "Physics-informed deep neural network for rigid-body protein docking." *MLDD workshop of ICLR 2022*. 2022.

**Questions:**

1. The paper mentions the absence of "co-evolution representation" derived from protein sequences. Could the authors provide a more detailed explanation of this point?
2. In the binding site prediction experiment, only a random split is compared, which is expected to perform worse than DiffMaSIF. Could additional baselines be included for a more comprehensive evaluation?

---

### Meta-Review · Area_Chair_nb3a · 2023-12-06

**Metareview:**

The paper tackles the important problem of predicting protein-protein complexes and the proposed diffusion-based approach is promising. Unfortunately the authors have not provided any feedback on important concerns raised in the reviews. We strongly encourage the authors to significantly revise their manuscript to provide more details on the approach and its evaluation,  improve experimental setup, and better justify the relevance of the proposed dataset.

**Justification For Why Not Higher Score:**

The paper has significant issues, as all reviews indicate. The authors did not respond to the reviewers.

**Justification For Why Not Lower Score:**

N/A

---

### Decision · Program_Chairs · 2024-01-16

Reject